# Comparison between Acupuncture and Nutraceutical Treatment with Migratens^®^ in Patients with Fibromyalgia Syndrome: A Prospective Randomized Clinical Trial

**DOI:** 10.3390/nu12030821

**Published:** 2020-03-19

**Authors:** Schweiger Vittorio, Secchettin Erica, Castellani Cinzia, Martini Alvise, Mazzocchi Elena, Picelli Alessandro, Polati Enrico, Donadello Katia, Valenti Maria Teresa, Dalle Carbonare Luca

**Affiliations:** 1Department of Surgery, Odontostomatology and Maternal Sciences, Fibromyalgia Diagnosis and Treatment Centre, University of Verona, 37134 Verona, Italy; alvise.martini@aovr.veneto.it (M.A.); enrico.polati@univr.it (P.E.); katia.donadello@univr.it (D.K.); 2Department of Surgery and Oncology, General and Pancreatic Surgery, Pancreas Institute, University of Verona Hospital Trust, 37134 Verona, Italy; 3Department of Medicine, Regional Specialized Centre for Biomolecular and Histomorphometric Research on Skeletal and Degenerative Diseases, University of Verona, 37134 Verona, Italy; cinziacats@yahoo.it (C.C.); elena.mazzocchi.vr@gmail.com (M.E.); mariateresa.valenti@univr.it (V.M.T.); luca.dallecarbonare@univr.it (D.C.L.); 4Department of Neurosciences, Biomedicine and Movement Sciences, Neuromotor and Cognitive Rehabilitation Research Sciences, University of Verona, 37134 Verona, Italy; alessandro.picelli@univr.it

**Keywords:** acupuncture, fibromyalgia, nutraceutical, dietary supplementation, quality of life, pain

## Abstract

Objectives: Fibromyalgia syndrome (FMS) is a chronic clinical condition characterized by pain, fatigue, altered sleep, and cognitive disturbances. The purpose of this study was to compare two alternative treatments (nutraceutical and acupuncture) in FMS patients through a randomized clinical trial. Research Methods: A total of 60 FMS female patients were randomized for treatment with a nutritional combination containing coenzyme Q10, vitamin D, alpha-lipoic acid, magnesium, and tryptophan (Migratens^®^ Group) or acupuncture treatment (Acupuncture Group) performed according the principles of traditional Chinese medicine (TCM), both for 3 months. Changes in pain and in quality of life (QoL) measured with a Fibromyalgia Impact Questionnaire Score-Revised (FIQ-R) and the Fibromyalgia Severity Scale (FSS) were performed at 1, 3, and 6 months after the start of treatments. Results: A total of 55 patient completed the study (21 in the Migratens^®^ Group and 34 in the Acupuncture Group). Migratens^®^ treatment shows a statistically significant reduction of pain 1 month after the start of therapy (T1, *p* = 0.025), strengthened after 3 months with maintenance of treatment (*p* = 0.012). The efficacy in reducing pain was apparent in the Acupuncture Group at all post-treatment determinations and at follow-up (T1 and T2 *p* = <0.001). Regarding QoL, improvement in FIQ-R and FSS values was revealed in both groups. Conclusion: The nutraceutical approach with Migratens^®^ seems to be an effective option to for patients with FMS. Our experience confirmed also the validity of acupuncture in these patients. Considering the complexity of the management of FMS patients, our results suggest a cyclical and sequential, or even concurrent treatment with different approaches, to improve the efficacy and the compliance of patients to long-term treatment

## 1. Introduction

Fibromyalgia syndrome (FMS) is a chronic clinical condition characterized by widespread pain, fatigue, unrefreshed and/or altered sleep, and cognitive disturbances [1]. FMS patients reported also several symptoms like muscle stiffness, irritable bowel syndrome (IBS), headache, temporomandibular joint dysfunction (TMJD), and others [2]. A recent literature review reported an overall prevalence of FMS in the general population worldwide between 0.2 and 6.6%, with a strong predominance in the female gender [3]. Patients with FMS reported a poor health status, one significantly lower compared to other chronic painful conditions widely accepted as disabling [4]. For this reason, the claims related to the disability of the FMS represent an emerging problem for health services in many countries [5]. According to the international guidelines, FMS treatment should be based on multimodal rehabilitation programs, and psychological and pharmacological treatments [6]. Furthermore, complementary and alternative treatments, including acupuncture and nutraceutical supplements, are used often as “add on” therapy in FMS patients [7]. To date, no pharmacological therapy has proven to be able to significantly improve pain, quality of life (QoL), or associated symptoms in FMS patients [8] and the role of complementary and alternative treatments has not yet been clearly evaluated for lack of large clinical studies [9]. Nevertheless, some of these approaches appear to be effective in the treatment of FMS [7]. In particular, since patients with FMS produce more damaging free radicals than healthy subjects, and they have a reduced antioxidant capacity, nutraceutical antioxidants could be an effective therapeutic approach in this setting, as suggested by previous observations [10]

The aim of this prospective, randomized controlled clinical trial was to evaluate the efficacy of acupuncture compared to nutraceutical supplementation with Migratens^®^ in a population of FMS patients referred to a FMS diagnosis and treatment center in Italy.

## 2. Methods

### 2.1. Participants

This prospective, randomized controlled clinical trial was conducted at the University Hospital of Verona, Department of Anesthesia and Intensive Care, FMS Diagnosis and Treatment Center and Department of Medicine, Regional Specialized Center for Biomolecular and Histomorphometric Research on Skeletal and Degenerative Diseases. All participants were informed about the details of the research and signed the informed consent document before the enrollment. The study was approved by the local Clinical Research Committee (FrIDAy protocol, ID 1917CESC) and registered in the Clinical Trial Registry (Clinicaltrial.gov ID: NCT04098757). The study complied with the revised ethical guidelines of the Declaration of Helsinki. The inclusion criteria were diagnosis of FMS from at least 6 months, according to the ACR (American College of Rheumatologist) diagnostic criteria 2016 (symptoms at the same intensity level for at least 3 months, widespread pain index ≥7 and symptom severity score ≥5 or widespread pain index 4–6 and symptom severity score ≥9, almost 4 of 5 body regions affected by pain) [11], female gender and age ≥18 years. Exclusion criteria were concomitant enrollment in other clinical studies, allergies to components of the nutritional preparation, chronic treatment with oral hypoglycemic agents and/or insulin, recent acupuncture treatments or acuphobia, severe psychiatric or neurological disorders, and current pregnancy. Patients who added other FMS pharmacological treatments during the observational period were excluded after enrollment. 

### 2.2. Interventions

Patients were randomized through sealed numbered envelopes and randomly assigned to one of the two intervention groups. We performed a simple randomization with a 1:1 allocation. In Group A FMS patients were scheduled for treatment with a nutritional supplement combination. The nutritional preparation was selected from the current literature (Pubmed, Uptodate, and Google Scholar) using the keywords fibromyalgia and nutraceutical treatment. The literature reported efficacy evidence in FMS for coenzyme Q10, vitamin D, alpha-lipoic acid, magnesium, and tryptophan [7,12,13,14,15]. Considering the nutraceutical composition and properties, the best therapeutic option was found to be a nutritional product containing all the substances considered above (Migratens^®^, Laborest srl, Assago, Milano, Italy). Migratens^®^ is classified as food supplement, and has been registered with the Italian Ministry of Health (aut. no. 86406). Furthermore, its fast-slow delivery technology is considered to be useful to prolong the therapeutic effects [16]. Migratens^®^ was prescribed regardless of patient’s concomitant pharmacological therapy (“add-on treatment”). The therapeutic protocol, according to the manufacturer’s instructions, was 1 sachet twice daily for 12 weeks. The two daily administrations had to be taken on an empty stomach by dissolving the product in a glass of water, spaced by a time interval of about 10 h. The nutraceutical product was provided for free by the manufacturer, stored in boxes with no identification of the contents. In the Group B, patients were scheduled for acupuncture treatment performed according the principles of traditional Chinese medicine (TCM). Briefly, acupuncture was always performed by the same licensed operator, following the principles of traditional Chinese medicine (TCM). According to TCM, diagnosis is based essentially on an accurate history and a careful analysis of wrists and tongue. Patients were classified in an excess condition (prevalence of humidity, stasis of qi or stasis of blood) or deficit condition (qi or yang spleen deficiency, yang spleen and kidney deficiency, blood deficiency of liver, liver and kidney yin deficiency). Excess situations were prevalent in the study population, although patients often had the symptomatology of two or more syndromic manifestations at the same time.

The most recurrent points among patients were: dumai (Governing Vessel) 20 and 24, heart 7, large intestine 4, renmai (Conception Vessel) 12 and 6, gall bladder 21 and 34, stomach 36, spleen 6, bladder 60, kidney 3, liver 3. These points were not performed in all patients in the same time, often some of these acupoints were combined with others to be more specific. In fact, the therapeutic TCM approach in fibromyalgia has two aims: to treat underlying conditions and to treat the area affected by pain (meridians involved), using local, adjacent, and remote points. In some cases, microsystems, in particular abdomino-puncture and auricolo-therapy, have been used as a support of somatic treatment or to reduce the number of needles used, also depending on the sensitivity of the patients. 

Therefore, the number and mode of needle insertions varied from case to case, but sometimes also from time to time, in the same subject per therapy session, made as personalized as possible, depending on the clinical response or changes in the wrists and tongue. 

The sought-after answer was the evocation of de qi or the perception of correct point grip by the acupuncturist.

Therefore, the choice of acupoints, the insertion depth, and the needles angulation were personalized, based on international criteria and influenced by therapeutic purpose, as well as patient’s constitution and pain sensitivity [17]. The acupuncture session, with a scheduled duration of at least 30 min, was performed by a specialist in internal medicine with an expertise in TCM. Acupuncture points were not the same in all patients and some acupoints were combined with other more specific ones. Therefore, the number and mode of needles inserted varied from case to case, but sometimes also from time to time, in the same subject per session. Needle stimulation was manual. The needles used were sterile, disposable, coated with copper wire, and handled with a guide. Each patient was scheduled for at least one course of acupuncture (2 sessions per week, for a total of 10 session), after an interval of at least 1 month. 

### 2.3. Data Collection

Socio-demographic and clinical data, including concomitant pharmacological therapy, were collected at baseline during the enrollment. Other variable data, including efficacy measures, were collected at baseline and at 1, 3, and 6 months. The 6-month interval was considered a follow-up determination to assess the durability of effects.

### 2.4. Efficacy Measures

The primary efficacy end-point was the change in pain measured with the visual analog scale (VAS 0–10) from baseline (T0) to 1 month (T1), 3 months (T3), and 6 months (T6, end of observation) in both treatment groups. The secondary efficacy end-point was the change in quality of life (QoL) measured with the Fibromyalgia Impact Questionnaire Score-Revised (FIQ-R) [18], validated for the Italian population [19], and Fibromyalgia Severity Scale (FSS, sum of the WPI and SSS) [20] from baseline (T0) to 1 month (T1), 3 months (T2), and 6 months (T3, end of observation) in both treatment groups.

### 2.5. Adverse Events

The type and frequency of adverse events (AEs) were registered and compared between two groups during treatments. The drug daily intake and the adherence of treatment was assessed through a self-reported diary. All the reasons leading to discontinuation of therapy were also registered.

### 2.6. Correlations 

As additional analysis, the potential determinants of pain intensity and disease severity were also investigated at baseline (T0) and at the end of treatment (T3). In particular, an exploratory data analysis was performed between VAS, FIQ-R, and FSS values and physical exercise advised for FMS patients (meditative physical activities like yoga or Tai Chi, for at least one hour 3 times a week or regular aerobic physical activity like walking, swimming, and cycling at least 30 min per day) [21]. The same correlation was investigated also in relation to depression level measured by the Hamilton Score for depression [22].

### 2.7. Statistical Analysis 

Statistical analysis was conducted with SPSS for windows 22.0 (IBM Corp Released 2016, NY, USA). Sample size was calculated using the statistical software PASS 14.0.8 (Kaysville, UT 84037 USA) by the “test for paired means”, considering a potency of 90% and an alpha error of 5%. Values were expressed as mean ± standard deviation. Based on the current literature about the efficacy of acupuncture in FMS measured with the mean VAS reduction [23], 23 patients for each treatment group was considered appropriate. Based on the literature and clinical experience regarding usual dropout of FMS patients (10%), a total of 60 patients was scheduled. Demographic, medical, and clinical characteristics were summarized by descriptive statistics using an independent sample *t*-test for continuous variables and a chi-square test for categorical variables. The VAS, FIQ-R, and FSS scores between time points were assessed with a paired sample *t*-test within each part of the study. The exploratory analysis to evaluate the correlations between VAS, FIQ-R, and FSS scores and physical exercise was performed through a linear regression. The effect size was calculated with G * POWER software.

## 3. Results 

### 3.1. Study Population

From December 2018 to October 2019, 60 female patients suffering from FMS were enrolled in the study according to the inclusion criteria, and randomized to receive Migratens^®^ treatment (Group A) or acupuncture treatment (Group B). Patient demographics and baseline characteristics are summarized in Table 1. Details on concomitant FMS drug consumption at baseline are reported in Figure 1. Despite randomization, median age between groups was statistically significant, with lower ages in patients of Group A (48.2 ± 7.4 vs. 52.9 ± 8.5, *p* = 0.04). Also, VAS values were statistically different between groups, with more pain severity in Group B (7.7 ± 1.7 vs. 8.5 ± 1.4, *p* = 0.04). In Group A, a total of 5 patients dropped out at follow-up (3 patients during the first month and 2 patients after the first follow-up observation, T1). In Group B, none of the included patients dropped out from acupuncture treatment and all reached the last follow up observation (Figure 2).

### 3.2. Efficacy

Regarding the primary end-point, in Group A (Migratens^®^ treatment), the change in VAS score was statistically significant at T1 (1 month after the start of treatment), ranging from 7.5 ± 1.7 at baseline to 6.6 ± 1.7 (*p* = 0.025) and also at T2 (3 months after the start of treatment) ranging from 7.3 ± 1.6 at baseline to 6.2 ± 2.9 (*p* = 0.012). In this group, the change in VAS score was not statistically significant at T3 (6 months after the start of treatment and 3 months after treatment interruption according to the study protocol), ranging from 7.3 ± 1.6 at baseline to 7.1 ± 2.2 (*p* = 0.6). In Group B, (acupuncture treatment) the change in VAS score was statistically significant at T1 (1 month after the start of treatment), ranging from 8.5 ± 1.4 at baseline to 6.4 ± 2.2 (*p* < 0.001) and at T2 (3 months after the start of treatment), ranging from 8.5 ± 1.4 at baseline to 6.6 ± 2.5 (*p* < 0.001). In this group, the change in VAS score was statistically significant also at T3 (6 months after start of treatment and 3 months after treatment interruption according to the study protocol), ranging from 8.5 ± 1.4 at baseline to 6.9 ± 2.4 (*p* < 0.001; Figure 3). Regarding secondary end-points, in Group A (Migratens^®^ treatment) the change in FIQ-R and FSS scores was not statistically significant at T1 (1 month after the start of treatment), ranging from 69 ± 15.9 at baseline to 64.5 ± 17.9 for FIQ-R (*p* = 0.2) and from 21.5 ± 5.2 at baseline to 20.1 ± 4.2 for FSS (*p* = 0.2), at T2 (3 months after the start of treatment), ranging from 69 ± 15.9 at baseline to 62.8 ± 20.5 for FIQ-R (*p* = 0.2) and from 21.5 ± 5.2 at baseline to 19 ± 7 for FSS (*p* = 0.3) and also at T3 (6 months after start of treatment and 3 months after treatment interruption according study protocol), ranging from 69 ± 15.9 at baseline to 66 ± 15.3 for FIQ-R (*p* = 0.5) and from 21.5 ± 5.2 at baseline to 19.5 ± 5.6 for FSS (*p* = 0.3). In Group B (acupuncture treatment) the change in FIQ-R and FSS scores was statistically significant at T1 (1 month after start of treatment), ranging from 74.2 ± 18.2 at baseline to 62.1 ± 3.7 for FIQ-R (*p* < 0.001) and from 23.4 ± 0.7 at baseline to 19.9 ± 0.9 for FSS (*p* = 0.001), at T2 (3 months after the start of treatment) ranging from 74.2 ± 3.1 at baseline to 59.4 ± 26.2 for FIQ-R (*p* < 0.001), and from 23.4 ± 0.7 at baseline to 19.6 ± 6.1 for FSS (*p* = 0.001) and also at T3 (6 months after start of treatment and 3 months after treatment interruption according study protocol) ranging from 74.2 ± 3.1 at baseline to 64.5 ± 25.3 for FIQ-R (*p* < 0.001) and from 23.4 ± 0.7 at baseline to 20.4 ± 6 for FSS (*p* = 0.003; Figure 4 and Figure 5). The comparison between the two groups showed that the reduction difference in primary and secondary end-points was greater in Group B (acupuncture) compared to Group A (Migratens^®^ treatment) at all the time intervals (Figure 6, Figure 7 and Figure 8).

Considering the results for VAS score in the two groups and the difference between T0 and T1 with a correlation of 0.5, the calculated effect size of group A was 0.53, while for group B it was 0.89.

### 3.3. Treatments Discontinuation

Overall, 5 patients discontinued treatments during observation, all in Group A (Migratens^®^ treatment). 4 patients discontinued for AEs, while 1 patient left for a personal medical decision related to an upcoming major surgical intervention. 

### 3.4. Safety

In the analyzed population, 11 patients (18.3%) reported AEs. Regarding the type of AEs, 6 patients in the Migratens^®^ Group (23.07%) reported gastrointestinal side-effects (diarrhea, nausea, dyspepsia, constipation, and lack of appetite). Of these, the 4 patients reporting gastrointestinal AEs discontinued the treatment with Migratens^®^. In the Acupuncture Group, 5 patients (14.7%) reported AEs related to treatment (small hematoma at the injection site, rapidly reabsorbed) but none left the treatment until the end of the study. 

### 3.5. Correlations 

Regarding potential determinants of pain intensity and disease severity, patients with aerobic physical activity at baseline (T0) showed better results in all the rating scales (VAS, FIQ-R, and FSS) compared to sedentary lifestyle patients (*p* < 0.05). A direct correlation was also observed at baseline between VAS/FIQ-R values and depression (*p* < 0.05). At T3, in Migratens^®^ Group, VAS, FIQ-R, and FSS values were not related to aerobic physical activity or depression (*p*> 0.1), suggesting that the benefit observed at the end of treatment is attributable only to the nutraceutical effects. In the Acupuncture Group, at T3, VAS, FIQ-R, and FSS values remains directly related to aerobic physical activity and levels of depression (*p* < 0.05; Figure 9). 

## 4. Discussion

### 4.1. Summary of Current Findings

Fibromyalgia treatment remains a major challenge for pain specialists. Pharmacological therapy showed discouraging results in most of the treated patients, who therefore discontinue the prescribed medications because of either a lack of efficacy and/or tolerability problems [24]. The most recent FMS treatment guidelines highlight the change in attitudes regarding the overall approach to FMS, in particular with regards to the use of pharmacological agents. According to these guidelines, pharmacological therapy should be considered only as an adjunctive treatment to non-pharmacological interventions in FMS [25]. In fact, the results from a UK online survey confirmed this statement, indicating that the range of mean pain efficacy was similar for pharmacological and non-pharmacological treatments, whereas non-pharmacological treatments had lower side effects and higher acceptability compared to pharmacological approaches [26]. For these reasons, we evaluated a nutraceutical approach in FMS patients with Migratens^®^, a composite preparation containing several substances (coenzyme Q10, vitamin D, alpha-lipoic acid, magnesium, tryptophan, niacin, and riboflavin) commonly singularly prescribed, in comparison with acupuncture therapy, a widely used treatment, in a randomized clinical trial. Our study demonstrated positive results for both treatments. In particular, regarding pain, Migratens^®^ treatment showed a statistically significant reduction 1 month after the start of therapy (T1, *p* = 0.025), strengthened after 3 months with maintenance of treatment (T2, *p* = 0.012), but diminished once the product was suspended (T3, *p* = 0.6). This data suggests that the clinical benefit observed with Migratens^®^ was attributable to the nutraceutical itself and cyclical or sequential regimes could be considered to improve compliance and adherence of patients to treatments, even if further studies are needed to validate this hypothesis. Our study suggests for the first time the efficacy of a combined nutraceutical approach in this specific setting. The results obtained in terms of VAS reduction are promising even if the improvement of other clinical parameters didn’t reach the statistical significance, probably due to less severe clinical expressions of the disease in this group at baseline. 

The efficacy in reducing pain was also highlighted in acupuncture treatment, with relevant improvement at all-time evaluations (T1 and T2 *p* = < 0.001), also significant even 3 months after the end of acupuncture cycle (T3, *p* < 0.001). Comparing the two groups, acupuncture showed greater reduction of pain at all-time intervals, while a more severe starting condition in this group compared to Migratens^®^ was highlighted. Regarding QoL evaluations, improvement in FIQ-R and FSS values was revealed in both groups, with statistically significance only in the Acupuncture Group (*p* < 0.001) though a clear improvement also emerged in the Migratens^®^ Group. Adverse events were limited, and only 4 patients of the Migratens^®^ Group discontinued the treatment due to gastrointestinal side effects. 

### 4.2. Nutrient Considerations

A nutraceutical approach in FMS treatment has been proposed based on several observations [7]. These studies note that FMS patients often demonstrated nutritional deficiencies and can therefore benefit from the intake of preparations containing vitamins, minerals, anti-oxidants, and amino acids [27]. The most widely used nutraceuticals products in FMS contain vitamin D and vitamin B complex, as well as minerals or essential amino acids. It has been proven that these nutrients contribute to normal cellular function at the central nervous system level [28]. The amino acid tryptophan is the precursor for the endogenous synthesis of serotonin and melatonin, which regulate the mood and sleep-wake cycle. Tryptophan deficiency has been described in FMS patients and its supplementation showed positive effects on sleep, mood, and fatigue [14,29,30]. The role of magnesium in FMS pathophysiology is still controversial, although it is reported that the magnesium deficiency can cause muscle cramps, weakness, fatigue, and asthenia [31]. Though there has been evidence that there is a low level of serum magnesium in FMS patients, a strong correlation with the severity of the FMS symptoms has not yet been demonstrated [8,32,33,34,35]. However, magnesium citrate treatment resulted in a significant reduction in pain symptoms and in the FIQ-R score [15]. A further improvement was observed in the degree of depression if magnesium was combined with amitriptyline treatment [15]. Alpha-lipoic acid (thioctic acid) is an organic acid characterized by strong antioxidant activities, both direct (protection from both intracellular and extracellular free radicals) and indirect (regeneration of other antioxidants such as vitamin C, vitamin E, coenzyme Q10, and Glutathione). Other activities, like the optimization of the oxidative metabolism of sugars and an enzymatic role within the Krebs cycle have also been described [36]. Alpha-lipoic acid treatment demonstrated clinical efficacy in various types of chronic pain, both neuropathic and inflammatory, acting both on inflammation (inhibiting the activation of nuclear factor kappa-light-chain-enhancer of activated B cells (NF-kB) and, therefore, of immune cells and microglia), and on the pain chronicization (anti-inflammatory action at the spinal level) [7]. In FMS patients, alpha-lipoic acid is evaluated both alone and in combination with pregabalin [37,38]. Coenzyme Q10 (or ubiquinone) is a compound involved in the transport of electrons in the mitochondria and in cellular oxidative phosphorylation. It acts as a free radical acceptor with antioxidant and membrane stabilizing properties. A reduction in coenzyme Q10 may lead to a deterioration of energy processes and a lower production of adenosine triphosphate (ATP) and is involved in several pathogenetic mechanisms [39]. Some evidence has highlighted the role of oxidative stress in producing pain and dysfunction in FMS patients [40]. In particular, low levels of coenzyme Q10 appear to increase the amount of mitochondrial superoxide and the level of lipid peroxidation of mononuclear cells in the blood of fibromyalgia patients [41]. FMS patients with coenzyme Q10 deficiency treated with coenzyme supplementation showed improvement in pain symptoms, fatigue, and sleep disturbances [13]. In addition, animal models proved that the coenzyme Q10 deficiency may lead to increased levels of pro-inflammatory cytokines (IL-1, IL-18) and abnormal activation of the NLRP346 inflammosome, which may counteract coenzyme Q10 supplementation [42]. Vitamin D is an important element in the prevention of musculoskeletal diseases throughout life [43,44]. Hypovitaminosis D is frequently detected in in osteo-articular and other chronic painful diseases, suggesting a correlation between these conditions and pain [45]. Moreover, a consistent correlation between vitamin D deficiency and depression and sleep disturbances has been highlighted [46,47]. Vitamin D supplementation seems to improve pain symptoms, quality of life, levels of depression, and sleep quality in patients with chronic widespread musculoskeletal pain [48]. In FMS patients, the role of vitamin D deficits and supplementation is controversial. In some systematic reviews, a correlation between vitamin D deficiency and fibromyalgia was determined, whereas its role in the pathophysiology and its clinical relevance requires further controlled studies [12,49]. More recent reviews reported as inconclusive the relationship between hypovitaminosis D and FMS, although vitamin D supplementation was considered as a co-adjuvant in FMS therapy [50]. Finally, although it is known that the absorption of B group vitamins like niacin and riboflavin is reduced by psychological factors, stress, and the use of sleeping-inducing drugs (which is common in FMS patients) [51], there are no studies in the literature about the efficacy of these supplementations in fibromyalgia patients. On the other hand, a recent study suggests the potential relationship between physical activity and vitamin B6 levels in the prevention of inflammatory articular processes [52]. Physical activity and vitamin B supplementation could be considered synergic approaches in the prevention of pain and potential inflammatory processes. 

### 4.3. Acupuncture Considerations

Conversely, the acupuncture approach in FMS is widely use in clinical practice. For this reason, acupuncture in FMS was evaluated in recent years by several studies. A recent systematic review and meta-analysis of randomized controlled trials reported that acupuncture therapy is an effective, safe, and recommended treatment for management of patients with FMS [53]. Other analyses have provided evidence that verum acupuncture was more effective than sham acupuncture for pain relief, improving sleep quality, and reforming general status in FMS post-treatment, although evidence that it reduces fatigue was not found [33]. There is also evidence that acupuncture potentiates the effect of antidepressants and exercise on pain by up to 30%, with poor, almost nil, adverse effects [17]. Regarding analgesic effects, verum acupuncture, rather than sham or placebo acupuncture, lead to changes in serum serotonin, neuropeptide Y and SP levels that may be contributing to long-term improvements on clinical outcomes in FM treatment [54,55]. In our study, the higher baseline VAS score associated to the most severe patients, usually more resistant to treatments, emphasized the efficacy of acupuncture in this setting. The mean duration of pain relief after a single cycle of acupuncture in FMS patients was estimated, in various meta-analyses, to be about 1 month, and it was not maintained at six months of follow-up [17]. Among complementary treatment in FMS, acupuncture is believed to be effective if compared with no or standard treatment, even if further high-quality trials are needed to investigate its benefits, harms, and mechanisms of action [56]. 

### 4.4. Study Limitations

To date, this is the first study on Migratens^®^ preparation in FMS patients. However, despite the relevant results highlighted, the study has also some limitations. The lack of a placebo control group, while methodologically acceptable even if with several practical difficulties, was not approved by local ethical committee, both treatments being currently considered valid in literature for the treatment of FMS and already in common clinical practice. Furthermore, although we have evaluated the symptomatic benefit of the treatments proposed in the two groups, the next step will be to evaluate the weekly or monthly clinical impact on the patient’s QoL with regard to painkiller reduction and the related side effects. Finally, in order to perform more significant considerations on the therapeutic role of acupuncture and Migratens^®^ in FMS patients, we are aware that the population sample will have to be implemented, as well as the time of observation. 

## 5. Conclusions

Due to the imbalance in the benefit/side effects relationship of the conventional pharmacological therapy, nutraceutical and acupuncture treatments seem to be effective therapeutic alternatives for the reduction of pain symptoms and for the improvement of QoL in FMS patients and could be considered in this setting. In particular, our study suggests that the nutraceutical approach with Migratens^®^, a composite product of several nutraceutical principles, seems to be an effective option for patients with FMS, with a low incidence of side effects. Our experience confirmed also the validity of acupuncture in these patients, as stated by the most recent literature. Considering the complexity of the management of FMS patients, our results can also provide a rational approach for polymodal treatment: concurrent, sequential and/or cyclical treatment with different approaches could improve efficacy and optimize the compliance of patients to long-term treatment.

## Figures and Tables

**Figure 1 nutrients-12-00821-f001:**
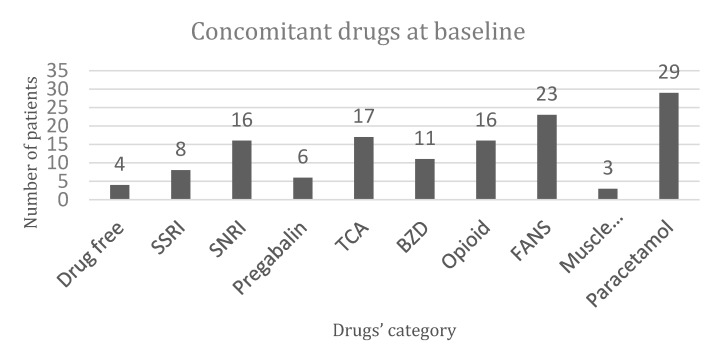
Concomitant pharmacological fibromyalgia syndrome (FMS) treatments in 60 eligible patients. SSRI = serotonin selective reuptake inhibitors; SNRI = serotonin noradrenaline selective inhibitors; GBPs = gabapentinoids; TCA = tricyclic antidepressants; BZD = benzodiazepines; OPI = opiates; NSAIDs = non-steroidal anti-inflammatory drugs; MR = muscle relaxants; ACT = acetaminophen.

**Figure 2 nutrients-12-00821-f002:**
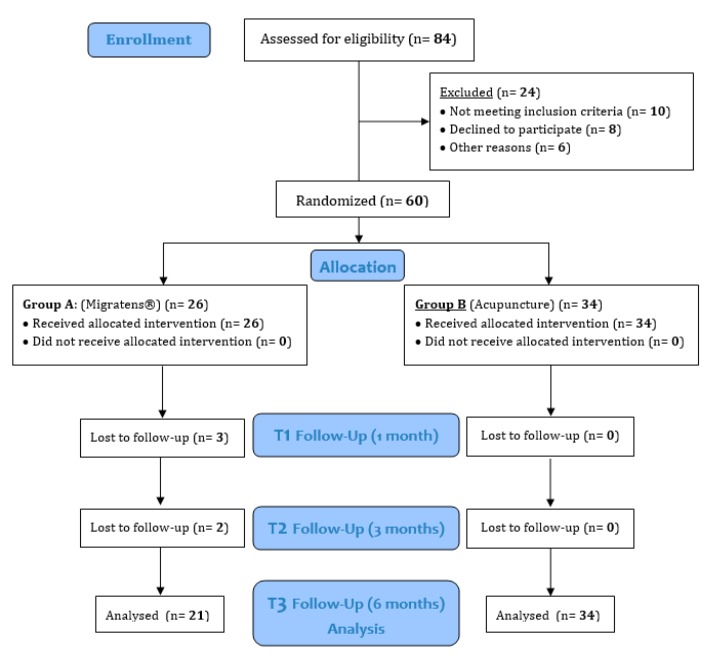
Study flow chart and follow-up.

**Figure 3 nutrients-12-00821-f003:**
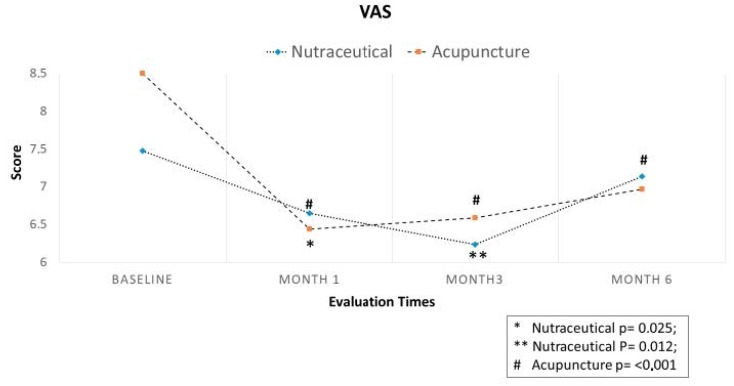
Primary efficacy end-point in analyzed populations: change in visual analog scale (VAS) pain score during observation. The score evaluation shows a statistically significantly pain intensity reduction at all the observations except in Migratens^®^ Group at T3 (6 months and 3 months after treatment interruption according study protocol, *p* = 0.6).

**Figure 4 nutrients-12-00821-f004:**
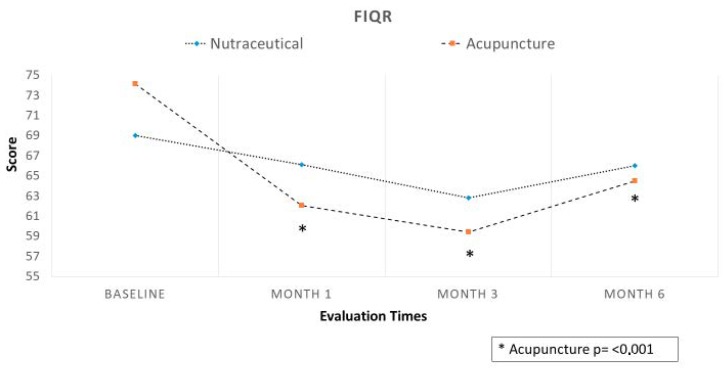
Secondary efficacy end-point in analyzed population: change in Fibromyalgia Impact Questionnaire Score-Revised (FIQ-R) score during observation. The evaluation shows a statistically significantly score reduction only in Acupuncture Group at all the observations (*p* < 0.001).

**Figure 5 nutrients-12-00821-f005:**
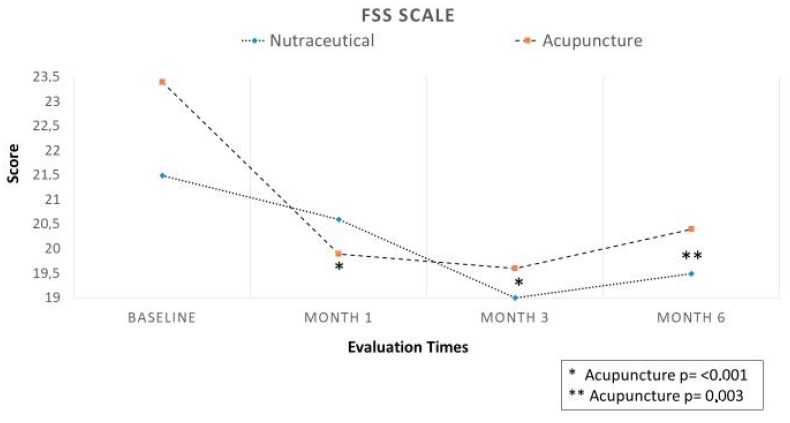
Secondary efficacy end-point in analyzed population: change in Fibromyalgia Severity Scale (FSS) score during observation. The evaluation shows a statistically significantly score reduction only in the Acupuncture Group in all the observations (*p* < 0.0001).

**Figure 6 nutrients-12-00821-f006:**
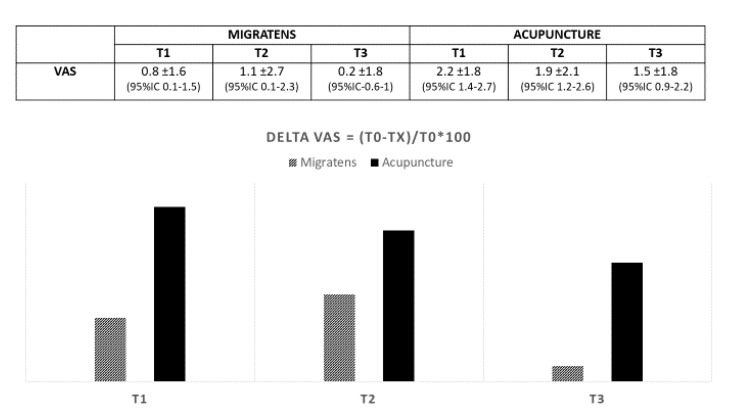
Reduction difference in VAS values in the two treatment groups during observation (values are expressed as mean difference ±SD).

**Figure 7 nutrients-12-00821-f007:**
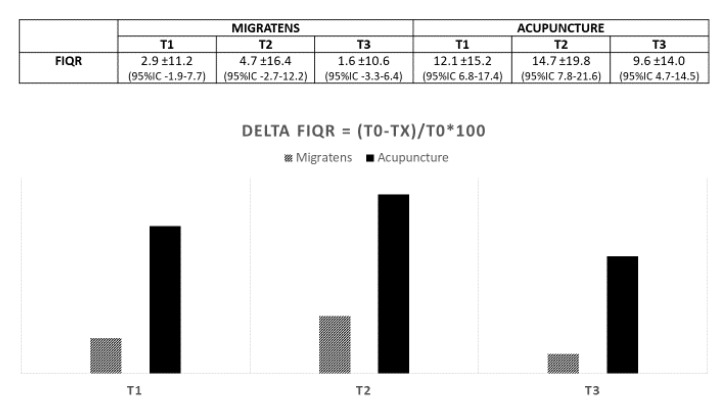
Reduction difference in FIQ-R values in the two treatment groups during observation (values are expressed as mean difference ± SD).

**Figure 8 nutrients-12-00821-f008:**
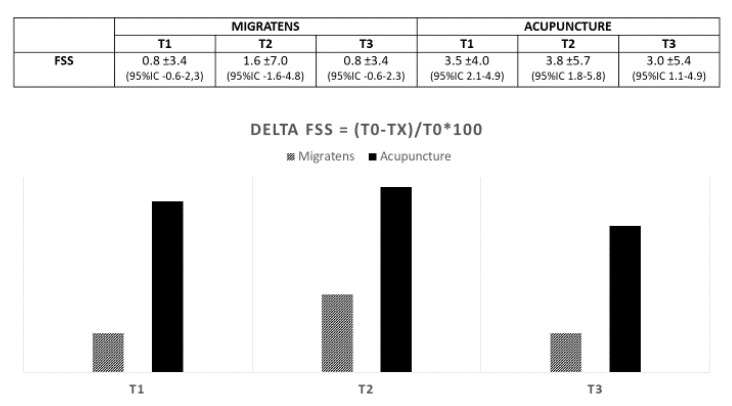
Reduction difference in FSS values in the two treatment groups during observations (values are expressed as mean difference ±SD).

**Figure 9 nutrients-12-00821-f009:**
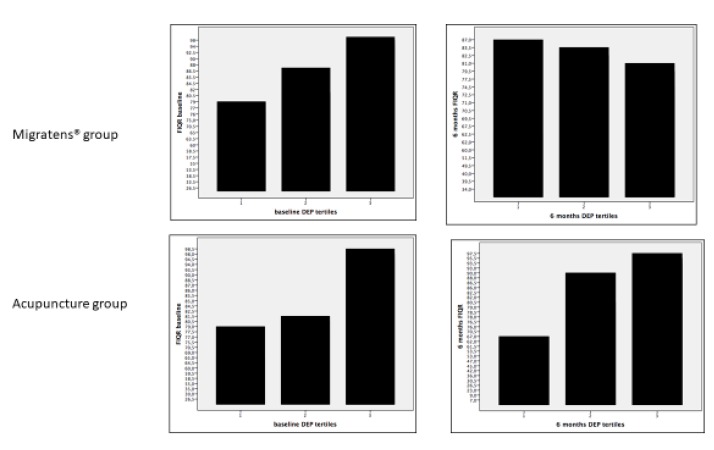
Linear correlation between FIQ-R and depression score expressed as tertiles in the two groups of treatment. Note that in Migratens^®^ Group, the positive correlation observed at baseline (*p* < 0.05) was lost at the end of follow-up, while it was maintained in the Acupuncture Group (*p* < 0.05).

**Table 1 nutrients-12-00821-t001:** Patient demographics and baseline characteristics.

Included in the Study	60 Patients		*p*
Treatment Groups	Group A (26)	Group B (34)	
Age, yr., mean (SD)	48.2 ± 7.4	52.9 ± 8.5	0.04
Pain (VAS 0–10)	7.7 ± 1.7	8.5 ± 1.4	0.04
FIQ-R (0–100)	69 ± 15.9	74.2 ± 18.2	ns
FSS (0–31)	21.5 ± 5.2	23.4 ± 4	ns

VAS = visual analoge scale; FIQ-R = Fibromyalgia Impact Questionnaire Score-Revised; FSS = Fibromyalgia Severity Scale.

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
