# Peer review of "Comparison between Acupuncture and Nutraceutical Treatment with Migratens® in Patients with Fibromyalgia Syndrome: A Prospective Randomized Clinical Trial"

_nutrients, 2020, doi:10.3390/nu12030821_

Round 1
Reviewer 1 Report
Very interesting and well writing paper about the effect of a nutraceutical intervention or acupuncture in FMS. Some minor aspect must be corrected or clarified:
INTRODUCTION
1. The introduction section is brief and complete. However, a brief paragraph that justified the use of antioxidants in fibromyalgia should be included.
METHODS
2. In order to allow the replication of the study, the authors should describe more precisely the applied acupuncture protocol (e.g. the main acupoints stimulated, the number of acupoints stimulated, the diagnosis of the patients on the basis of the Traditional Chinese Medicine, etc.). Tabulated specific data of each patient or general data of the acupuncture protocol should be included (as supplemental file).
3. Did the authors use the validated version of the FIQ-R and FSS for the Italian population? Please, include the validation articles of each questionnaire for the Italian population.
4. Migratens® is under the control of some Drug Association like FDA or L´Agenzia Italia del Farmaco (AIFA)? Please, provide this information in the text.
5. The patients were classified taking into account the physical activity level. How the authors did this classification? Have they used any tool like the International Physical Activity Questionnaire?
RESULTS
6. Figures 6, 7 and 8 have redundant information (tables and graphics display almost the same data). Because graphics show less information than tables, graphics should be removed.
DISCUSION
7. At baseline, patients from Group B (acupuncture) have greater VAS scores than patients from Group A (Migratens®). Did this fact determine the observed results in the VAS change scores in Group B?
CONCLUSIONS
8. The conclusion section is too conclusive. The absence of a control group in the study design makes the results more doubtful.
Line 343: I recommend the use of “seems to be effective” instead “effective”
Line 345: Replace “demonstrate” by “suggest”
Author Response
Referee 1
Comments and Suggestions for Authors
Very interesting and well writing paper about the effect of a nutraceutical intervention or acupuncture in FMS.
We are thankful to the referee for the globally favourable comments on our manuscript. We’ll do our best to improve it according to the referee’s suggestions.
Some minor aspect must be corrected or clarified:
INTRODUCTION
- The introduction section is brief and complete. However, a brief paragraph that justified the use of antioxidants in fibromyalgia should be included.
It’s a good point. We added in the introduction a paragraph explaining the relationship between fibromyalgia and antioxidants.
METHODS
- In order to allow the replication of the study, the authors should describe more precisely the applied acupuncture protocol (e.g. the main acupoints stimulated, the number of acupoints stimulated, the diagnosis of the patients on the basis of the Traditional Chinese Medicine, etc.).
We detailed more precisely the applied acupuncture protocol, as follows:
Acupuncture was performed always by the same licensed operator, following the principles of Traditional Chinese Medicine (TCM). According to which, diagnosis was based essentially on an accurate history and a careful analysis of wrists and tongue. Patients were classified in an excess condition (prevalence of humidity, stasis of qi or stasis of blood) or deficit condition (Qi or Yang Spleen deficiency, yang Spleen and Kidney deficiency, blood deficiency of Liver, Liver and Kidney yin deficiency). Excess situations were prevalent in the study population, although patients often had the symptomatology of two or more syndromic manifestations at the same time.
The most recurrent points among patients were: Dumai 20 and 24, Heart 7, Large Intestine 4, Renmai 12 and 6, Gall bladder 21 and 34, Stomach 36, Spleen 6, Bladder 60, Kidney 3, Liver 3. These points were not performed in all patients in the same time; often some of these acupoints were combined with others more specific. In fact, therapeutic TCM approach in fibromyalgia has two aims: to treat underlying condition and to treat the area affected by pain (meridians involved), using local, adjacent and remote points. In some cases, microsystems, in particular abdomino-puncture and auricolo-therapy, have been used as a support of somatic treatment or to reduce the number of needles used, also depending on the sensitivity of the patients.
Therefore the number and mode of needle insertions varied from case to case, but sometimes also from time to time in the same subject per therapy session, made as personalized as possible, depending on the clinical response or changes in the wrists and tongue.
The sought-after answer was the evocation of de qi or the perception of correct point grip by the acupuncturist.
Tabulated specific data of each patient or general data of the acupuncture protocol should be included (as supplemental file).
As specified above, we reported in the text general data of the acupuncture protocol. However, considering the number of sessions performed and the variability of treatments between patients or in the same patient during the therapeutic cycle (depending on the clinical response), we felt that to insert a summary table would likely to be confusing and complex.
- Did the authors use the validated version of the FIQ-R and FSS for the Italian population? Please, include the validation articles of each questionnaire for the Italian population.
The modified version (FIQ-R) translated and validated in Italian language was used for this study.
We included the follow validation article in the references:
Salaffi F., Franchignoni F., Giordano A. et al. Psychometric characteristics of the Italian version of the revised Fibromyalgia Impact Questionnaire using classical test theory and Rasch analysis. Clin Exp Rheumatol 2013; 31 (Suppl. 79): S41-S49.
- Migratens® is under the control of some Drug Association like FDA or L´Agenzia Italia del Farmaco (AIFA)? Please, provide this information in the text.
Since Migratens is classified as food supplement, it was notified to the Italian Ministry of Health (aut. no. 86406). We added this sentence in the text.
- The patients were classified taking into account the physical activity level. How the authors did this classification? Have they used any tool like the International Physical Activity Questionnaire?
To evaluate physical activity in patients, we did not use the International Physical Activity Questionnaire, but we considered as an adequate physical activity a walking of at least 30 minutes per day or practicing meditative physical activities like Yoga or Tai Chi or postural gymnastics for at least one hour three times a week. We specified this point in the correlation section.
RESULTS
- Figures 6, 7 and 8 have redundant information (tables and graphics display almost the same data). Because graphics show less information than tables, graphics should be removed.
In our opinion graphs have a greater visual impact for readers and could contribute to highlight the results. However, if for the reviewer are redundant, we can remove it.
DISCUSSION
- At baseline, patients from Group B (acupuncture) have greater VAS scores than patients from Group A (Migratens®). Did this fact determine the observed results in the VAS change scores in Group B?
We agree. The higher baseline VAS score associated to most severe patients, usually more refractory to treatments, emphasized the efficacy of acupuncture in this setting. On the other hand, the less severe clinical expressions of the disease in group A at baseline could explain the non-significant improvement in some clinical parameters. We added this observation in the text.
CONCLUSIONS
- The conclusion section is too conclusive. The absence of a control group in the study design makes the results more doubtful.
Line 343: I recommend the use of “seems to be effective” instead “effective”
Line 345: Replace “demonstrate” by “suggest”
We mitigated the conclusions of the study changing the sentences, as suggested.

Reviewer 2 Report
This manuscript describes a study in which fibromyalgia subjects underwent treatment regimens of a nutritional supplement (containing coenzyme Q, vitamin D, alpha-lipoic acid, magnesium, and tryptophan) or acupuncture (personalized regiment, according to Traditional Chinese Medicine principles) for 3 months, with a follow-up at 6 months. Both treatments produced benefits, with those produced by acupuncture being more sustained over time.
The study is well-designed, well-referenced, and reports interesting and important results. The manuscript would benefit from some minor adjustments, and a few edits.
- The Abstract should identify the components of the nutraceutical. This is done elsewhere, but it would be helpful if they were also in the Abstract.
- There is no placebo group in this study, for ethical and understandable reasons. The authors should calculate effect sizes for the two treatments, as this will provide a further indicator of efficacy that can be compared to other treatments (both pharmacological and non-pharmacological).
- Re Randomization. Lines 76-78 seem to indicate block randomization. If so, how did the groups end up with the final numbers that they did? With dropouts in the nutrient group, the end numbers are uneven.
- The 6-month interval is a follow-up determination to assess durability of effects. Use this terminology, and include it in the abstract. The information is there, but it can be phrased more clearly.
- Figs 3-5. The placement of the asterisks does not seem to correspond to stats reported in the Results. It looks like both red and blue points should have asterisks in some cases, while in F4 the red acupuncture points should have them. Please review the placement of asterisks to indicate significance in these two figures. At the moment, it is not clear what the asterisks in these figures mean. They do not correspond to what is stated in the text.
- Discussion. This is very dense. Create a new paragraph on line 262 with “A nutraceutical approach in FMS treatment has been proposed based on several observations.” (note some word adjustment) Then another new paragraph line 314, with “The acupuncture treatment..” This will make P1=summary of current findings, P2=nutrient considerations, P3=acupuncture considerations.
- Re cyclical and sequential suggestion. Why not concurrent? The two methods seem to address different aspects of dysfunction (nutritional, integrative), and so provide a rational approach for polymodal treatment. Perhaps this could be mentioned. Given the promising results reported in this study, I certainly hope that the authors can perform a follow=up study. And if you do, make sure you include effect sizes in your analysis.
Minor edits
Line 29: was apparent in the…. at all post-treatment determinations and at follow-up (be explicit)
Line 34: suggest a cyclical and sequential, or even concurrent, (see comment 6 above)
Line 250: but diminished once the product
Line 252: could be considered to (see comment 6)
Line 263: These studies note that FMS patients
Line 269: Tryptophan deficiency has been described
Line 273: has not yet been demonstrated
Lines 318, 322: there is no need for quotation marks around the word verum
Line 324: that may be contributing to (there may well be other contributors, one only sees what one addresses)
Line 327: Among complementary
Line 337: Daily impact may be too often, consider weekly (one can encounter questionnaire burnout if this is too frequent; also, with an expectation of day-to-day variations, the value of daily is not clear)
Author Response
Referee 2
Comments and Suggestions for Authors
This manuscript describes a study in which fibromyalgia subjects underwent treatment regimens of a nutritional supplement (containing coenzyme Q, vitamin D, alpha-lipoic acid, magnesium, and tryptophan) or acupuncture (personalized regiment, according to Traditional Chinese Medicine principles) for 3 months, with a follow-up at 6 months. Both treatments produced benefits, with those produced by acupuncture being more sustained over time.
The study is well-designed, well-referenced, and reports interesting and important results. The manuscript would benefit from some minor adjustments, and a few edits.
We are thankful to the referee for the globally favourable comments on our manuscript. We’ll do our best to improve it according to the referee’s suggestions.
- The Abstract should identify the components of the nutraceutical. This is done elsewhere, but it would be helpful if they were also in the Abstract.
It’s a good point. We added in the abstract the components of the nutraceutical.
- There is no placebo group in this study, for ethical and understandable reasons. The authors should calculate effect sizes for the two treatments, as this will provide a further indicator of efficacy that can be compared to other treatments (both pharmacological and non-pharmacological).
Considering the results for VAS score in the two groups and the difference between T0 and T1 with a correlation of 0.5, the calculated effect size of group A was 0.53, while for group B was 0.89 (The effect size was calculated with G*POWER software). We added this information in the text.
- Re Randomization. Lines 76-78 seem to indicate block randomization. If so, how did the groups end up with the final numbers that they did? With dropouts in the nutrient group, the end numbers are uneven.
Sample size calculations for the primary outcome, with 90% power and two-sided 5% significance, showed that we needed to include 46 participants. The calculation was performed with PASS program (14.0.8).
Considering a rate of drop out about 10% we finally needed to include 52 patients, 26 for each arm.
We performed a simple randomization with a 1:1 allocation. We specified this information in the statistical section and corrected the description of the randomization.
- The 6-month interval is a follow-up determination to assess durability of effects. Use this terminology, and include it in the abstract. The information is there, but it can be phrased more clearly.
As suggested, we specified the role of the 6-month interval in the method session.
- Figs 3-5. The placement of the asterisks does not seem to correspond to stats reported in the Results. It looks like both red and blue points should have asterisks in some cases, while in F4 the red acupuncture points should have them. Please review the placement of asterisks to indicate significance in these two figures. At the moment, it is not clear what the asterisks in these figures mean. They do not correspond to what is stated in the text.
We checked the placement of asterisks in the abovementioned figures and modified the label between the two groups to ameliorate the readability of the figures.
- Discussion. This is very dense. Create a new paragraph on line 262 with “A nutraceutical approach in FMS treatment has been proposed based on several observations.” (note some word adjustment) Then another new paragraph line 314, with “The acupuncture treatment..” This will make P1=summary of current findings, P2=nutrient considerations, P3=acupuncture considerations.
Thank you for the suggestion. We reorganize the discussion dividing in three different sections our comments.
Re cyclical and sequential suggestion. Why not concurrent? The two methods seem to address different aspects of dysfunction (nutritional, integrative), and so provide a rational approach for polymodal treatment. Perhaps this could be mentioned. Given the promising results reported in this study, I certainly hope that the authors can perform a follow=up study. And if you do, make sure you include effect sizes in your analysis.
Thank you for the suggestion. We agree with the rationale of a polymodal treatment. We mentioned this point in the conclusion.
Minor edits
Line 29: was apparent in the…. at all post-treatment determinations and at follow-up (be explicit)
Line 34: suggest a cyclical and sequential, or even concurrent, (see comment 6 above)
Line 250: but diminished once the product
Line 252: could be considered to (see comment 6)
Line 263: These studies note that FMS patients
Line 269: Tryptophan deficiency has been described
Line 273: has not yet been demonstrated
Lines 318, 322: there is no need for quotation marks around the word verum
Line 324: that may be contributing to (there may well be other contributors, one only sees what one addresses)
Line 327: Among complementary
Line 337: Daily impact may be too often, consider weekly (one can encounter questionnaire burnout if this is too frequent; also, with an expectation of day-to-day variations, the value of daily is not clear)
Thank you for these specific comments that further improve the impact of the paper. We corrected all the points suggested in the text.
